# Self-Reported Prevalence of HIV Infection, Sexually Transmitted Infections and Risky Sexual Behavior among Mental Health Care Users Accessing Healthcare Services in Tshwane District, South Africa

**DOI:** 10.3390/healthcare9101398

**Published:** 2021-10-19

**Authors:** Mathildah Mpata Mokgatle

**Affiliations:** 1Department of Biostatistics, School of Public Health, Sefako Makgatho Health Sciences University, Ga-Rankuwa 0208, South Africa; mathildah.mokgatle@smu.ac.za; 2School of Transdisciplinary Research and Graduate Studies, College of Graduate Studies, University of South Africa (UNISA), Muckleneuk, Pretoria 0001, South Africa

**Keywords:** HIV, STI, risky sexual behaviors, MHCU, HIV risk perception, South Africa

## Abstract

The rate of HIV and sexually transmitted infections among mental patients is higher than that of the general population worldwide. Many risky sexual behaviors are associated with mental illness. However, mental health care users (MHCUs) are not specifically targeted for HIV preventative care, and routine HIV testing is not done among this population. Limited studies have investigated self-reported HIV and STI prevalence and associated risky sexual behavior in persons with mental illnesses accessing health care services in South Africa in particular. This study set out to determine both the Sexually Transmitted Infections (STI) and human immunodeficiency virus (HIV) self-reported prevalence and sexual practices of MHCUs. A descriptive cross-sectional survey using purposive sampling was used to select 107 MHCUs across five clinics within Gauteng Province of South Africa who were above the age of 18, had a mental illness, and were currently stable and receiving chronic medication. Descriptive statistics were performed using Stata IC version 16. The chi square test was used to indicate statistical significance (*p* < 0.05) of differences in frequency distributions. More males (52.5%) than females were currently in a sexual relationship (50.0%), having multiple sexual partners (*n* = 4.13%), and having alcohol-driven sex (*n* = 4.19%). The majority of MHCUs (*n* = 82.77%) had an STI in the past six months, and a quarter (*n* = 21.25%) were HIV-positive with over two-thirds of MHCU (*n* = 69.70%) perceiving themselves not at risk for HIV. MHCUs engaged in risky behaviors had a low perception of the risks of contracting HIV. Bivariate analysis of gender by sexual behavior revealed that female MHCUs are more at risk of being forced to have sex compared to males (*p* = 0.006). Integrated interventions should be put in place to ensure that MHCUs’ sexual and reproductive health are not left behind and issues such as sexual education, safe sex, and sexually transmitted infections should form part of the care of MHCUs.

## 1. Introduction

Sexually transmitted infections (STIs) remain highly prevalent worldwide and they increase an individual’s susceptibility for HIV infection, making this a major public health concern [1]. Globally there were 37.7 million people living with HIV in 2020, and among the South African population, the total number of people living with HIV (PLWHA) was estimated to be approximately 8.2 million in 2021 [2]. Mental disorders have been found to be related to the risk of sexually transmitted diseases with elevated rates of high risk sexual behavior among this population group [3,4,5]. Currently, severe mental disorders account for approximately 12% of the global burden of disease and were predicted to reach 15% by 2020 [5]. Mental illness in itself presents with symptoms such as low impulse control, impaired judgment and reality testing, instabilities in affect as well as increased sexual activity during illness episodes, which may increase the sexual risk behavior and risk for HIV infection [4]. Mental health care users (MHCUs) are therefore a vulnerable population and treating them as nonsexual increases their risk of HIV infection [5,6]. Studies done in sub-Saharan Africa over time have shown a higher HIV prevalence among MHCUs than in the general population, and prevalence was also found to be higher among women than in men [7,8]. Despite developing countries having the highest prevalence of HIV infection in the world, much lower HIV testing rates have been reported [9,10]. Not only do these findings show a lack of targeted and regular HIV testing among MHCUs as a preventative strategy in developing countries, but also the majority of MHCUs in high endemic developing countries could be living with undiagnosed and untreated HIV infection. In addition, sexual risk behaviors such as having multiple sexual partners, engaging in unprotected sex, having sex in exchange for money and drugs, and having sex with prostitutes were found to be contributing to the high prevalence of HIV among this population [4,9]. The psychological problems that affect personal cognition, emotional regulation, and behavior are some of the reasons for this association [11].

Given that mental illness is a risk factor for HIV, it is necessary for research to examine both the STI and HIV prevalence and sexual practices among this population [3,4]. Available research among this group informs us that health services are not comprehensive as they are specialized, and limited to clinical psychiatric health; hence, the sexual and reproductive health of MHCUs is not included or prioritized.

The study thus aims to determine both the STI and HIV self-reported prevalence and sexual practices of mental health care users, and will contribute to the body of knowledge in developing comprehensive, inclusive, and integrated programmes for the MHCUs. Knowing the extent of the sexual risk and vulnerability of the MHCUs towards STI/HIV will alert the health care providers to effectively manage mental illness and HIV comorbidity among this population and improve their health [9,11,12,13].

## 2. Materials and Methods

### 2.1. Study Design and Population

The study setting for this cross-sectional design consisted of five clinics within a sub-district of Tshwane District in Gauteng Province of South Africa. The sub-district has a total of 18 clinics and only five clinics offer chronic mental health services, due to the scarcity of psychiatric clinicians that are supposed to implement and integrate mental health services in the primary health care system [14]. A convenient sample of all the five clinics was included in the study in order to have access to the MHCUs registered at those clinics. Adult mental health care users attending mental health clinics in the sub-district made up the study population. Only mental health care users who had been in the mental health programme for more than one year were included in the study. Participants were stable, on medication, cognitively alert, and capable of giving informed consent. With a population size of *n* = 800, we treated the five clinics within Tshwane sub-district 6 as a unit to ensure the representativeness of the data proportional to the number of MHCUs. The minimum sample was *n* = 51 per clinic using Raosoft^®^ (Online Sample Size Calculator; Raosoft, Inc., Seattle, WA, USA) for sample size calculation at 95% confidence level and 5% margin of error. The total estimated sample size in all the five clinics was *n* = 250 MHCUs. Purposive sampling was used to select MHCUs above the age of 18, having a mental illness, and being currently stable and receiving medication at the clinic.

### 2.2. Measures

An assessment tool adapted from the mini-mental status examination was used to assess or screen participants’ readiness to take part in the study [15]. Participants who exhibited adequate cognitive capacity were included. Data were then collected through a structured researcher-administered questionnaire. The data collection tool included socio-demographic, alcohol, and substance use questions. Some of the sexual behavior questions focused on having a current sexual partner, number of sexual partners, having been treated for STIs in the past six months, notifying a partner of the STI symptoms, ever had unplanned or spontaneous sex, use of condom during sex, consistent use of condoms, ever had unwanted or forced sex, ever had sex under the influence of alcohol, and ever had sex in exchange for money. HIV risk questions included risk perception of HIV infection, ever had HIV counselling and testing, awareness of sexual partner’s HIV status, and ever received HIV prevention education

### 2.3. Data Collection

The data were collected using a researcher-administered standardized questionnaire in English. The questionnaires were administered by trained research assistants. The questionnaire was developed by referring to previous tools and constructs obtained from the review of the literature on STIs and risky sexual behaviors [16]. The MHCUs were requested to participate immediately after they had completed the session with the clinician to ensure accurate screening for cognitive capacity. The tool was administered on a one-on-one basis in a private room that was provided by the clinic to ensure privacy and confidentiality. The period of data collection was June 2018 to February 2019.

### 2.4. Data Analysis

The analysis of the data was done using the STATA IC version 16.0 statistical package (STATA Corp., College, TX, USA). Descriptive statistics such as frequencies, percentages, and proportions were computed to describe the study variables. Socio-demographic and risky behavioral characteristics were compared by sex. The Pearson Chi-square was used to examine the differences between male and female participants regarding demographic and behavioral characteristics and the sexual characteristics of the MHCUs. Statistical significance was set at *p* < 0.05 for all variables.

### 2.5. Ethical Considerations

The Research Ethics Committee of Sefako Makgatho Health Sciences University (SMUREC/H/284/2015: IR) reviewed the protocol and gave an ethical clearance. Permission was granted by the Tshwane District Health, the selected health care facility managers, and the mental health care programme managers, who are the immediate custodians of the patients in the Mental Health Unit. Assessment or screening of the mental health readiness was done by the data collector, who is a mental health nurse, before recruiting the participants into the study. Informed consent was obtained from all MHCUs. Participation was voluntary, including the right to withdraw from the study without any preconditions. For anonymity purposes, no identifying information was collected and the data file was password-protected, with access limited to the lead investigator.

## 3. Results

### 3.1. Demographic and Behavioural Characteristics of Participants

Instead of the estimated sample size of *n* = 250, the final sample size obtained was *n* = 107 patients, yielding a response rate of 43%. The low response rate was due to the institutional refusal to grant permission for accessing the patients during the Life Esidimeni crisis inquest [17,18]. Health care facility managers took the decision to protect the MHCUs accessing care in their institution as they were sensitised by the allegations of neglect of MHCUs during the crisis period.

Half (55.4%) of the study participants were males. The mean (±SD) age of the participants was 43 (±11.6) years. Greater than one-third (43.9%) of the participants were over the age category of 45 years. The majority (74.8%) of the participants were single, and 7.5% were employed (Table 1). One-third (37.4%) of MHCUs use substances with the preferred substance of choice being tobacco (75.0%).

### 3.2. Risky Sexual Behaviours of MHCUs

With respect to risky sexual behavior, more males (52.5%) than females were currently in a sexual relationship (50.0%), more males reported multiple sexual partners (*n* = 4, 13%) and having had alcohol driven sex (*n* = 4, 19.1%), while more females reported forced sex (*n =* 8, 16.7%). Equally two-thirds of both males and females reported use of condoms; however, more males were comfortable in demanding the use of a condom (58.8% vs. 55.6%). An overwhelming majority of females had tested for HIV in the last 12 months (89.6% vs. 79.7%), and more males knew their partner’s HIV status (61.3% vs. 54.2%). More males reported having an STI in the past six months (82.8% vs. 70.8%). Despite the majority of MHCUs having had an STI in the past six months (82, 77.3%), over two-thirds of MHCUs perceived themselves not at risk for HIV (Table 2).

An overwhelming majority (77.3%) reported having an STI in the six months preceding the study, with a quarter (25.0%) of MHCUs being HIV-positive at the time of data collection.

## 4. Discussion

This study has described the sociodemographic, sexual behavioral characteristics and self-reported STI and HIV status of MHCUs accessing care in primary health care facilities in Gauteng, Tshwane District. The Gauteng Department of Health (GDoH) began the process of deinstitutionalization in the 1990s that promoted following the patients from the institution to the communities where they live by including multidisciplinary psychiatric teams in selected PHC facilities [19,20].

The study found a high overall prevalence of self-reported STI (77.4%) among the MHCUs. Very few studies have addressed the occurrence of STIs other than HIV in mentally ill patients, with those conducted earlier revealing that the prevalence of STI including HIV infection among the mentally ill was higher than that of the general population [8,20,21,22,23]. Most researchers do indicate, however, that various factors such as psychosocial, behavioral and environmental factors underlie the increased rate of HIV infection among mentally ill patients. Mental illness can increase the risk of other sexually transmitted infections, henceforth they can be attributed to the increased self-reported STI prevalence in this particular study. Furthermore, findings from a multi-site study of patients receiving treatment for serious mental illness confirm that STIs other than HIV are a major health problem [21].

The data showed that, compared to women, more men were currently in sexual relationships, had multiple sexual partners, and had alcohol-driven sex. Over time, evidence of dangerous sexual behaviors between men and women has been cited in the literature. Lundberg et al. [24] found a high presence of multiple sexual partners among men with a mental illness compared to women, and a higher association between high-risk sexual behaviors and being HIV-infected among men. Contradictory findings from previous studies, however, attributed a higher risk of HIV infection from high risk behavior to women [7,25,26]. In line with general population statistics, females reported being more frequently forced to engage in sexual acts as opposed to their counterparts. But this phenomenon is widely reported to being more prevalent in mentally ill females than those without psychiatric conditions. A similar finding was reported in a qualitative study examining sexual risk behavior and sexual abuse among MHCUs in Uganda that found that more women had experienced physical violence and abuse, with some reporting having been chased away from their homes during illness episodes, which exposed them to increased vulnerability related to homelessness [25].

Studies examining gender differences of sexual risk behavior reported a high prevalence of recent unprotected sex among both genders, which was slightly higher among women [10]. In addition, more women have been reported having engaged in unprotected sex in the previous six months compared to men, attributable to condom refusal by their partners, whereas more men reported having engaged in sex under the influence of alcohol and drugs compared to women. But these findings are contrary to the findings of this study, where men, although by slight margins, reported not using condoms (56.0% vs. 55.8%) and demanding condoms more than females (58.8% vs. 55.6%).

Despite the high prevalence of self-reported STI (77.1%), MHCUs in the study had a low HIV risk perception (28.0%). This finding is consistent with studies that showed the presence of low risk perception for HIV infection among MHCUs, particularly men, with this resulting in risky sexual behavior, poor health seeking behavior and low HIV testing levels [10]. Over half (58.2%) of MHCUs reported knowing their partner HIV status, and this finding is in stark contrast to studies that report a significant proportion of participants reporting not knowing the HIV status of their sexual partners, with inconsistent or no condom use being high among this population [3]. MHCUs in the study had a high HIV testing rate, with the majority (84.1%) having tested for HIV in the 12 months preceding the study. Moosa and Jeenah [9] argue that, given the high prevalence of HIV in South Africa, the widely reported presence of sexual risk behaviors among MHCUs and the accessibility of HIV testing in public health institutions, routine provider-initiated HIV testing of MHCUs is justified. Similarly, Joska et al. [26] assert that mandatory provider-initiated testing of all new psychiatric admissions is not only a necessary screening and assessment process, but also in the best interest of MHCUs. If HIV testing and counselling can be integrated into mental health services, it could be used to inform the MHCUs about the risks and benefits of HIV testing, which could also contribute to increasing the HIV testing rates among MHCUs.

The use of substances as demonstrated by other studies is associated with risky sexual behavior. Furthermore, in these studies, substance use is a strong predictor for STI and more so for patients with mental illness [6]. Although our study only reports on the proportion of substance use, over a third (37.4%) of MHCUs reported the use of substances with studies showing associations between substance use and STI and explain this association as a result of poor access to prevention care and impaired judgement [27,28].

## 5. Study Limitations

The results obtained in this study should not be generalized to all MHCUs. Less than 50% of the sample size was reached; hence, the study cannot be generalized to the larger population of the MHCUs. The STI infections were self-reported, and there is a possibility that STIs/HIV and risky sexual behaviors were under-reported. Although we limited the recall period to 12 months, participants might have had difficulties in recalling some events that happened in the past, and by doing so may have introduced recall bias. There is a possibility that mental health influenced the validity of recall in this study. However, during data collection, we assured the MHCUs about the anonymity of the study and findings.

## 6. Conclusions

The majority of MHCUs in this study engaged in sexual behavior that places them at risk for infection with HIV and other STIs. It is concerning to find up to 57% of the MHCUs reported to be in a sexual relationship and more than two-thirds reported to have had STIs. This suggests that they are predisposed to having casual sex and/or unprotected sex, since more than two-thirds reported to be single and less than half of the sample reported to be using condoms during sexual intercourse. The reported HIV prevalence is more than double the national prevalence. Data also confirms low STI partner notification in this group. This implies that routine screening for STI and HIV should be integrated into mental health programmes. Inclusion of care in the programmes should emphasize screening for substance use and casual sex to achieve sexual health of the MHCUs. Interventions should contain widespread sexual and reproductive health awareness on issues such as sexual education, safe sex, and sexually transmitted infections. In addition, the causal sexual relationships and the interplay between risky sexual behavior and mental health disorder should be examined.

## Figures and Tables

**Table 1 healthcare-09-01398-t001:** Distribution of sociodemographic and behavioral characteristics of mental health care users (*n* = 107).

Sociodemographic of Mental Health Care Users		
Characteristic		Frequency (%)
Sex	Female	48 (44.9)
	Male	59 (55.4)
Age category	22–35	29 (27.1)
	36–45	31 (29.0)
	>45	47 (43.9)
Level of study	Primary	69 (64.5)
	Secondary	38 (35.5)
Marital status	Single	80 (74.8)
	Ever married	27 (25.2)
Employment status	Employed	8 (7.8)
	Unemployed	95 (92.2)
Use of substance	Yes	40 (37.4)
	No	67 (62.6)
Type of substance	Alcohol	4 (10.0)
	Dagga	6 (15.0)
	Tobacco	30 (75.0)

**Table 2 healthcare-09-01398-t002:** Sexual characteristics of mental health care users.

Sexual Characteristics of Mental Health Users						
	Female		Male		*p* Value	
	Yes *n* (%)	No *n* (%)	Yes *n* (%)	No *n* (%)		Proportion (%)
Sexual relations						
Sexually active	24 (50.0)	24 (50.0)	31 (52.5)	28 (47.7)	0.794	51.3
Had more than one sexual partner	0(0.0)	24(100.)	4 (13.3)	26 (86.7)	0.063	7.4
Been forced to have sex	8 (16.7)	40 (83.3)	1 (1.7)	58 (98.3)	0.006 *	8.4
Alcohol-driven sex	0 (0.0)	6 (100.0)	4 (19.1)	17 (80.9)	0.247	3.7
Condom use						
Use condoms	25 (65.8)	13 (34.2)	35 (66.0)	18 (34.0)	0.980	65.9
Able to demand condom	20 (55.6)	16 (44.4)	30 (58.8)	21(41.2)	0.761	57.5
HIV testing and STI						
Has been tested for HIV in the last 12 months	43 (89.6)	5 (10.4)	47 (79.7)	12 (20.3)	0.163	84.1
Know partner’s HIV status	13 (54.2)	11 (45.8)	19 (61.3)	12 (38.7)	0.487	58.2
Had an STI in the last six months	34 (70.8)	14 (29.2)	48 (82.8)	10 (17.24)	0.144	77.4
HIV perceptions						
Know about HIV	46 (95.8)	2 (4.2)	57 (96.6)	2 (3.4)	0.833	96.3
At risk for HIV	15 (31.9)	32 (68.1)	15 (28.9)	37 (71.1)	0.740	28.0

* significant at *p* < 0.05.

## Data Availability

A dataset will be submitted upon request by the Editor.

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
