# Peer review of "Self-Reported Prevalence of HIV Infection, Sexually Transmitted Infections and Risky Sexual Behavior among Mental Health Care Users Accessing Healthcare Services in Tshwane District, South Africa"

_healthcare, 2021, doi:10.3390/healthcare9101398_

Round 1

Reviewer 1 Report

Addressing sexual and reproductive health among patients receiving mental health care is paramount. The subject is topical and could draw a great deal of interest to readers. However, the article requires extensive revision of content and presentation. Please see attached comments for your consideration.

Author Response

Dear reviewer

Reviewer 2 Report

Review for healthcare-1400430 - Self-reported prevalence of HIV infection, sexually transmitted infections and risky sexual behaviour among mental health care users accessing healthcare services in Tshwane District, South-Africa

GENERAL COMMENT FOR THE STUDY

The proposed study addresses an important public health worldwide concern that has been neglected over the years. The authors describe some issues related to HIV infection and other sexually transmitted infections in people with mental disorders.

The study appears to be well-designed, but there are several signs of weakness.

Therefore, in my assessment, the manuscript contains some limitations, which prevent its publication in the current format.

In general, authors should use original bibliographic sources (references) to write Introduction and discuss the results. For example, there are several data or citations mentioned in the manuscript which are in disagreement with the cited reference.

The author must standardize the number of decimal places (decimal digits) in all percentages presented in the text.

The details for each section are described below.

ABSTRACT

The author should inform and present its results regarding the hypothesis test (associations verified by Chi-square test).

Line 23: there is a missing “(“ before “n = 4, 13%)”.

INTRODUCTION

There is a missing reference in line 35.

MATERIALS AND METHODS

Line 90: “were” instead of “will be”.

Although the author informs that the convenience sampling (purposive sampling) was used, it would be interesting to inform what was the criterion adopted for the selection of the 5 clinics, because on page 3 (lines 138-140) she informs that "the low response rate was due to the institutional refusal rate to granting permission for accessing the patients during the Life Esidimeni crisis inquest [20]. Health care facility managers took the decision to protect the MHCUs accessing care in their institution."

There seems to be a contradiction here: why did the author choose locations where she knew of the prior existence of difficulty in accessing eligible participants for the study?

“2.2 Measures” and “2.3 Data Collection”

I suggest describing in more detail which variables were collected and which time period these data refer to.

“2.4 Data Analysis”

For the type of analysis performed, it makes no sense to define the dependent and independent variables. This definition is only needed in multivariate analysis or regression models.

Anyway, why was the variable sex chosen as the dependent variable? Are there previous studies that suggest the choice of this variable as an outcome that guides the bivariate analysis in relation to other dependent variables?

It makes more sense to define the two central themes of the study as the outcome: HIV infection and other sexually transmitted infections.

RESULTS

I suggest standardizing the use of n versus N.

Why didn't the author describe it in the text, discuss it in the discussion section, or include the variable level of study in the association analysis (table 2)?

This variable is a social determinant of health enshrined in the literature and should be better explored.

Why did the author not describe in the text the results of the association analysis (Chi-square test) contained in table 2?

Line 152: there is a missing “%”.

Suggestion for lines 156-157: Despite the majority of MCHU’s having had an STI in the past six months (82, 77.3%), over two thirds of MCHU perceived themselves not at risk for HIV (69, 70%) (Table 2).

Figures 1 and 2 can be removed.

DISCUSSION

Line 176: it would be interesting if the author explained how the provision of services in primary health care in the study area is, because in many health systems, clinics or mental disorders care facilities are in specialized care and not in primary care.

There are several data or citations mentioned in the manuscript which are in disagreement with the cited reference.

“Few studies have addressed the occurrence of STIs other than HIV in mentally ill patients, as the earlier study conducted in Brazil reported STI including HIV infection among the mentally ill being estimated to be at least eight times the prevalence in the general population [21].”

There is no study carried out in Brazil that showed this data and the reference #21 is not a study performed in Brazil.

Additionally, throughout the discussion, I identified some excerpts very similar to those contained in the paper "Vanable et al. (2008). Differences in HIV-Related Knowledge, Attitudes, and Behavior Among Psychiatric Outpatients with and Without a History of a Sexually Transmitted Infection."

I didn't find “Lundberg et al (2015)” (line 191) in the References section (pages 7-9).

I checked reference “Lundberg et al (2012)” (ref. #15) and theirs results differ from those presented by the author in the discussion (lines 191-193).

Author Response

Dear Reviewer

Round 2

Reviewer 1 Report

Looks better, thanks for taking the time to address the comments suggested. A few typos for your attention in the attached. Otherwise, I think it will be good to go.

Reviewer 2 Report

I have no further comments.

Author Response

Reviewer has no further comments and suggestions